Saliva as a diagnostic tool in soccer: a scoping review

Ferreira Joyce 1
Jimenez Manuel 2
Cerqueira Agatha 1
Rodrigues da Silva Joana 3
Souza Bruno 4
Berard Lucas 5
Bachi Andre L.L. 4
Dame-Teixeira Naile 3
Coto Neide 5
Heller Debora debora_heller@hotmail.com 1 6 7
1 Post-Graduate Program in Dentistry, Universidade Cruzeiro do Sul , Sao Paulo , Brazil
2 Department of Physical Education and Health, International University of La Rioja , Logrono , Spain
3 Post-Graduate Program in Dentistry, Universidade de Brasília , Brasilia , Brazil
4 Post-Graduate Program in Health Science, Santo Amaro University (UNISA), Santo Amaro, Brazil , Sao Paulo , Brazil
5 Post-Graduate Program in Dentistry, Universidade de São Paulo , Sao Paulo , Brazil
6 Hospital Israelita Albert Einstein , Sao Paulo , Brazil
7 Department of Periodontology, University of Texas Health Science Center at San Antonio , San Antonio , United States of America
Pawar Ajinkya
Electronic publication date: 2024 Oct 14
Publication date: 2024
Volume: 12
Electronic Location ID: e18032
Received 2024 Apr 15; Accepted 2024 Aug 12
Copyright: ©2024 Ferreira et al.
Copyright year: 2024
Copyright holder: Ferreira et al.
License: This is an open access article distributed under the terms of the Creative Commons Attribution License, which permits unrestricted use, distribution, reproduction and adaptation in any medium and for any purpose provided that it is properly attributed. For attribution, the original author(s), title, publication source (PeerJ) and either DOI or URL of the article must be cited.
License URL: https://creativecommons.org/licenses/by/4.0/

Keywords: Saliva, Salivary proteins and peptides, Soccer, Biomarkers, Sports medicine

Funding: The authors received no funding for this work.

==============================
Background

A high-performance sport like soccer requires training strategies that aim to reach peak performance at the right time for the desired competitions. Thus, the investigation of biochemical markers in saliva is a tool that is beginning to be used in athletes within the physical training process. There is still no evidence on universal saliva collection and analysis protocols in soccer. This review aims to map the use of saliva as a tool for analyzing athletic performance in soccer, from the biomarkers used to the validated protocols for these analyses.

Methods

A broad systematic literature search was carried out in the electronic databases Web of Science, Livivo, Scopus, PubMed, LILACS and gray literature (Google Scholar and ProQuest). Two reviewers selected the studies and extracted data on the type of salivary collection used, the salivary biomarker evaluated and monitored.

Results

Ninety-three articles were included. The most frequently analyzed salivary biomarkers were cortisol (n = 53), testosterone (n = 35), secretory immunoglobulin A (SIgA) (n = 33), salivary alpha amylase (n = 7), genetic polymorphisms (n = 4) and miRNAs (n = 2). The results of the studies indicated beneficial effects in monitoring salivary biomarkers in the assessment of sports performance, although most studies did not include a control group capable of comparison. Salivary collection and analysis protocols were varied and commonly not reported.

Conclusions

This scoping review provides a comprehensive overview of the current landscape of salivary biomarker research in soccer. The findings underscore the importance of these biomarkers in assessing athletes’ physiological responses and overall well-being. Future research should focus on refining methodologies, exploring additional biomarkers, and investigating the practical implications of salivary biomarker monitoring in soccer and other sports.

Introduction

Saliva is increasingly recognized as a valuable biofluid for diagnosing, monitoring, and predicting disease status. Its complexity, housing biomolecules like DNA, mRNA, microRNA (miRNA), proteins, metabolites, and microorganisms, positions saliva as an informative medium (Toman et al., 2022). The simplicity of saliva collection, its non-invasiveness, and cost-effectiveness contribute to its potential in biomedical research, personalized medicine, and health monitoring, offering specific and personalized biomarkers for precise disease treatment and diagnosis (González et al., 2008).

In sports medicine, the growing interest in salivary biomarkers reflects a pursuit to assess health-related aspects influenced by physical activity. Factors such as training status, fatigue levels, exercise type, intensity, duration, age, and gender can impact biomarker concentrations, with applications ranging from health monitoring to doping control in research (Palacios et al., 2015; Silva et al., 2020; Schönfelder et al., 2016). This expanding field underscores the potential of salivary analysis to provide tailored insights into individual health, contributing to a nuanced understanding of the effects of regular physical activity on the body. Saliva provides an easy-to-collect, relatively stress-free and non-invasive means of assessing athletes in various settings, including the field, in a practical way and does not need of specific medical training/ nor medical personnel (Casanova et al., 2015; Gatti & De Palo, 2011; Hayes et al., 2015), offering advantages over blood-derived biomarkers in terms of accessibility and athlete preference. However, significant limitations are the lack of specific reference values for physically active individuals, leading to challenges in establishing personalized reference scales (Borchers et al., 2022), and the lack of standardization of methods to collect and laboratory assessments (Mortazavi, Yousefi-Koma & Yousefi-Koma, 2024). Nevertheless, salivary analysis is gaining interest in talent identification, athlete development, and high-performance programs (Jiménez et al., 2020; Mendoza et al., 2021).

Given the above, the use of biomarkers in soccer, the most popular and consequently, most studied sport in the world has proven to be a promising tool for advancing and enhancing this sport (Cadegiani, Kater & Gazola, 2019). Furthermore, soccer’s global prevalence, the intensity and variety of physical exertion it requires, and the standardization of its play make it an exemplary sport for investigating salivary biomarkers, with the potential to impact a wide array of sports and contribute significantly to sports science and medicine.

The measurement of biomarkers, such as hormonal, protein, metabolic, and genetic markers, allows for a more accurate assessment of physical performance, muscle recovery, and athletes’ overall health. Furthermore, the identification and monitoring of specific biomarkers can aid in the early detection of injuries, the diagnosis of sports-related illnesses, and the implementation of personalized training strategies. For these analyses to be used in daily training, they must be routinely and repeatedly performed.

A high-performance sport like soccer requires training strategies aimed at reaching peak performance at the right time for targeted competitions. Thus, investigating biochemical markers in saliva is a tool that is beginning to be used in athletes within a physical training process. However, there is still no evidence regarding universal protocols for saliva collection and analysis in soccer. In this regard, it becomes necessary to map the use of saliva as a tool for analyzing athletic performance in soccer, from the biomarkers used to the validated protocols for these analyses. Therefore, here we aimed to map the literature on using saliva as a diagnostic tool in soccer, analyzing which salivary biomarkers are employed and describing the available protocols. This comprehensive scoping review is intended for sports scientists, performance analysts, soccer coaches, athletic trainers, and sports medicine professionals who are interested in integrating salivary biomarker analysis into their assessment and monitoring of athletes’ physiological responses and training outcomes.

Methods

Study design

A scoping review was conducted using the Preferred Reporting Items for Systematic Reviews and Meta-Analyses (extension for scoping reviews) checklist (Tricco et al., 2018).

Search strategy

A systematic search was carried out in the electronic databases Web of Science, Livivo, Scopus, PubMed, LILACS and gray literature (Google Scholar and Pro-Quest). General controlled vocabulary, MeSH terms and keywords were chosen (“athletes”, “Football”, “Soccers”, “saliva”, “oral fluids”, “salivary proteins and peptides” “diagnostic salivary”, “sports medicine”). The search strategy, adapted to each database, is detailed as follows: (athlet(es)*[Title/Abstract]) OR (football*[Title/Abstract]) OR (soccers*[Title/Abstract]) OR (saliva*[Title/Abstract]) OR (“oral fluids”[Title/Abstract]) OR (“salivary proteins or peptides”[Title/Abstract]) OR (“diagnostic salivary”[Title/Abstract]) OR (“sport medicine”[Title/Abstract]).

Randomised or non-randomised clinical trials and case reports were included. There were no language or year limitations. Duplicates were identified using EndNote Web (Clarivate Analytics, Mumbai, India) and then manually identified.

Eligibility criteria

Studies were included when the target population met the following criteria: exclusively investigating the presence of salivary biomarkers in soccer players. Studies with individuals of all ages, both sexes, professional athletes, or amateurs were included. Studies that did not involve “soccer” or “football” as the sport, were not original studies, were conference abstracts, or were written in a non-Latin alphabet that could not be translated online using tools such as Google Translate were excluded.

Selection of studies

A 2-step selection process was performed by two independent reviewers (J.F.P.P. and J.R.S.): (1) evaluation of all titles and abstracts retrieved and (2) reading of full-texts. Conflicts were resolved by consensus with a third reviewer (D.H.). Reference lists of selected articles were analyzed manually when searching for other eligible titles. The Rayyan QCRI web application tool (Qatar Computing Research Institute) was used to select the studies.

Data extraction and synthesis

The data was extracted by two independent reviewers, J.F.P.P. and J.R.S., in two stages. Firstly, J.F.P.P. and J.R.S., the two reviewers, assessed the eligibility of all titles and abstracts found through the open access tool Rayyan QCRI from the Qatar Computing Research Institute. Afterward, they conducted conflict evaluations between themselves. After completing the initial stage, data was gathered on the publication date, author, gender and age of the athletes evaluated, manner of saliva sampling and processing, and the salivary biomarkers studied. Subsequently, an evaluator (D.H.) re-evaluated each selected study and critically assessed the main information obtained.

Results

Characterization of the studies

The literature search was carried out using the descriptors and keywords in all the databases (Table 1). In phase one, after eliminating duplicates, 338 records were screened. From this sample, titles and abstracts were read, considering inclusion and exclusion criteria, resulting in the selection of 149 articles. A total of 189 studies were excluded as they focused on different sports other than soccer or were not related to sports. In phase two of article selection, the full text of these 149 articles was reviewed, resulting in the removal of 56 articles that did not meet the inclusion and exclusion criteria, leaving a final sample of 93 articles (Fig. 1). Table 2 presents an overview of the evaluated articles, followed by an individual qualitative analysis.

Table 1 Distribution of the number of articles found in each database.

Database	Total records found	
PubMed	212	
Proquest	20	
Google Scholar	41	
Lilacs	94	
Web of Science	65	
Total records	432	

Figure 1 Flow diagram of the study.

Table 2 Overview of included studies focused on salivary biomarkers.

First author, year, country, study type	Participantes gender and age	Sample size	Results focusing on salivary biomarkers	Saliva collection and processing methods	Salivary biomarkers assessed	
Akimoto, T. 2003 Japan Clinical trial	Female, Male 17 and 18 years old	21 players	Decrease in exercise-induced salivary IgA and increase in salivary cortisol were inhibited by acupuncture.	Stimulation of saliva secretion was achieved by chewing on a sterile cotton swab (Salivette, Sersted, Germany) at a frequency of 120⋅ 120 s−1. Saliva was separated from the cotton by centrifugation at 3,000 rpm. After measuring the sample volume, the saliva samples were frozen at −80 °C and stored until analysis.	Secretory Immunoglobulin A (sIgA) Cortisol	
Alzahrani, M. 2019 Sweden Clinical trial	Male	20 Players	No significant variation in the concentration of T was observed between pre and post-training, while DHEA significantly increased after short-duration exercises	The saliva was collected using Salivette. The stored samples were thawed and then centrifuged at 1000 × g for 5 min, and 0.5 mL of the supernatant was withdrawn.	Testosterone	
Alzharani, M. 2020 Saudi Arabia Observational study	Male age 20.6 ± 1.4 years, body mass 70.2 ± 1.6 kg, height 178 ± 2 cm, and BMI 22.2 ± 0.5 kg/m2	26 Players	PCA for saliva samples produces a less clear separation between pre and post-exercise samples.	For saliva sample collection, a tube was used; the samples were centrifuged at 1000 × g for 20 min at 4 °C; then, the inlay was removed from the tubes, and the samples were stored at −80 °C..	Adenosine triphosphate (ATP) Adenosine diphosphate (ADP) Creatine kinase	
Amato, A. 2018 Italy Clinical trial	Male	21 players	Soccer players, who have a superior strength-related TGS compared to power-related TGS.	Saliva sample was collected in a sterile 10 ml tube from each participant. Each sample was stored in the freezer.	Polymorphisms: ACE; ACTN3; CKMM; PPAR α; PPAR γC1; NRF2A/G; NRF2 C/T.	
Arruda, A. 2015 Brazil Clinical trial	Male 15 to 17 years old	39 players	Changes in testosterone and peak power changes were not related.	Samples of unstimulated saliva were collected through passive drool into sterile 15 ml centrifuge tubes for a period of 5 min. The saliva samples were then stored at −80 °C.	Testosterone Cortisol	
Arruda, A. F. 2016 Brazil Observational study	Male age, 19.3 ± 0.7 years	21 players	The playing venue did not cause a significant change in the concentration of salivary steroid hormones at rest.	There are no reports.	Testosterone	
Askari, B. 2011 Iran Clinical trial	Male average age of 21 ± 2 years	22 players	The secretion rate of IgA and the value of the s-IgA/total protein ratio showed significant reductions at the same time	Unstimulated saliva was collected over a 4-minute period.	Alpha-amylase Cortisol	
Azarbayjani, M. A. 2011 Iran Observational study	Male 10 to 14 years	9 Players	No significant difference in total protein concentration was observed. Although changes in salivary cortisol, α-amylase, and total proteins were observed concurrently, there was no significant relationship between them.	Saliva samples were collected from each player 30 min and 5 min before the start of the competition, at halftime, and again 5 and 30 min after the end of the competition	Testosterone Dehydroepiandrosterone	
Baldari, C. 2009 Italy Clinical trial	Male 10 to 14 years	51 players	The salivary concentrations of DHEAS were associated with standing long jump performance. Additionally, salivary concentrations of testosterone could not be related to leg explosive strength.	There are no reports.	Astaxanthin (Asx) sIgA	
Baralic, I. 2015 Serbia Observational study	Male	40 players	The supplementation of Asx improves the response of sIgA and attenuates muscle damage, thus preventing inflammation induced by rigorous physical training.	Unstimulated whole saliva was collected using Salivettes (Sarstedt, Nümbrecht, Germany), placing the swab under the tongue. Immediately after collection, the saliva samples obtained were separated from the cotton by centrifugation at 1500 ×g for 15 min. The supernatant fluid was frozen at −80 °C for subsequent sIgA analysis.	sIgA	
Bishop, N. 1999 United Kingdom Clinical trial	Male	8 players	The pattern of change in plasma cortisol, circulating lymphocyte count, and salivary immunoglobulin A secretion did not differ between carbohydrate and placebo trials.	Each participant was asked to swallow to empty the mouth before the saliva swab (Sarstedt, N mbrecht). The saliva swabs were then placed in their plastic containers and stored frozen at −70 °C until analysis.	Crystalline (characteristics of oral fluid at different periods of the training and competitive cycle).	
Bonato, M 2020 Italy Observational study	Male Age: 26 ± 6 years	17 players	It was observed that GE and GC showed similar concentrations of salivary cortisol at the beginning of the study and in response to a specific night soccer training session.	Saliva samples (Salivette, Sarsted AG & CO, Nümbrecht, Germany) were collected for salivary cortisol assessment (salivary cortisol kit, DiaMetra, DKO020) before (PRE), after (POST), and before sleep (POST 1) the SSG session.	Cortisol	
Broodryk, A 2020 South Africa Clinical Trial	Female 22 years old	43 players	The post-AFT (fatigue test) results showed a positive relationship in salivary cortisol (r = 0.3, p = 0.04), between ISP-anger with maximum heart rate (r = 0.3, p = 0.03), ISP-anger and YYIR level, as well as ISP fatigue (r = 0.4, p = 0.04), and between perceived effort rate and ISP vigor (r =  − 0.4, p = 0.008) as well as ISP fatigue (r = 0.3, p = 0.05).	Saliva samples were collected through passive drool for cortisol assessment. If necessary, participants could chew a piece of Parafilm™ to stimulate saliva flow. The saliva was then collected through a plastic straw into a 20 mL collection tube, after which the sample was stored in a refrigerator (at 4 ± 1 °C) and transported to a qualified laboratory for analysis.	Cortisol	
Broodryk, A 2017 South Africa Observational Study	Female 22 years old	47 players	The results indicated an increase in cortisol, psychological fatigue, and TMD from baseline and/or pre to post-FAT (fatigue test) (p < 0.05). Cortisol and RPE (rating of perceived exertion) (r =  − 0.34, p = 0.03) showed correlation post-FAT.	Saliva flow was stimulated by chewing on a piece of Parafilm, if needed, and collected through a plastic straw into a 20 ml collection tube. The samples were then stored at 4 ± 1°C, after which they were sent for analysis.	Cortisol	
Broodryk, A. 2021 USA Observational Study	Female	8 players	The cortisol correlated with TMD and various mood subscales before a winning outcome, with the Incredibly Short Mood Profile (ISP) correlating at all times with anxiety scores (p < 0.05).	Saliva collected 1 h before and 15 min after each game.	Cortisol	
Casto, K. V, 2016 USA Clinical trial	Female 18 and 22 years old	25 players	The levels of testosterone (T) and cortisol (C) increased during the competition but decreased in the 30 min following the end of the game.	The participants were given a piece of sugar-free gum (Trident®, original flavor) to stimulate saliva production.	Cortisol Testosterone	
Castro-Sepulveda, M 2018 Chile Clinical trial	Male Age: 16.8 ± 0.4 years	17 players	No differences were found before the matches in T (p = 0.38), C (p = 66), nor T:C (p = 0.38) between the groups.	The players remained seated, with eyes open, head slightly tilted forward, and making minimal orofacial movements. All saliva (± 3 ml) was collected for about 2 min. Saliva samples were centrifuged at 1,500 g for 15 min and stored at −20 °C until analysis.	Testosterone Cortisol	
D’Ercole, S.;  2013 Italy Cross-sectional study	12 anos	 40 players	The training period significantly decreased the concentration of SIgA.	There are no reports	Polymorphisms ACTN3 R577X, AMPD1 C34T, I/D ACE e M235T AGT	
Rodrigues de Araujo, V. 2018 Brazil Clinical trial	Male	32 players	The HIIE protocol (High-Intensity Interval Exercise) as a physical test conducted in soccer athletes increased the salivary concentration of exercise intensity markers such as lactate, total protein, but did not affect IgA levels.	Saliva samples were collected at rest and immediately after the HIIE protocol.	Testosterone Cortisol	
Di Luigi, L 2006 Italy Cross-sectional study	Male 13 years	110 players	The resting salivary cortisol concentration was positively correlated with chronological age (p < 0.01) and negatively correlated with % body fat (p < 0.05), while significant negative correlations of DeltasC and DeltasC% after exercise with age, pubertal stage, and mean testicular volume (p < 0.05) were observed. The post-exercise increase in salivary testosterone (sT) was correlated with chronological age, pubertal stage, and pre-exercise sT levels (p < 0.01), while DeltasT and DeltasT% increases were negatively correlated with chronological age and resting sT levels (p < 0.05 and p < 0.01, respectively).	All volunteers collected a saliva sample using a swab and a saliva collection tube (Salivette).	Cortisol, Testosterone, and Estradiol	
Dionísio, T 2017 Brazil Cross-sectional study	Male	220 players	Athletes with genotypes RR/RX (ACTN3) and DD (ACE) showed better performance in jump and run tests. On the other hand, athletes with genotype ID/II showed better results during endurance tests, while AGT genotypes did not seem to favor athletes during the assessed physical tests.	There are no reports	Cortisol Testosterone	
Edwards, D 2015 Observational study USA	Female Male		Women with relatively low levels of cortisol showed greater increases in testosterone during the competition than women with higher levels of cortisol.	The participants were given a piece of sugar-free gum (Trident®, original flavor) to stimulate saliva production, and a 20 ml polypropylene container that they were asked to fill up to a line marked at 5 ml on the side.	Testosterone Cortisol	
Edwards, D.2006 USA Clinical Trial	Female Male	22 players	For men, changes in T related to the game were positively correlated with the variables (p < 0.08), but T before the game was not. For women, T before the game was positively related to each of these variables (t(11) = 2.7, p < 0.03), but changes in T related to the game were not.	Participants were given sugar-free gum (Trident, original flavor) to stimulate saliva production and a 20 ml plastic vial that they were asked to fill up to a marked line at 5 ml on the side.	Testosterone	
Edwards, D. 2013 USA Observational study	Female	90 players	The salivary levels of cortisol and testosterone decreased in teammates who did not play but watched the competition from the sidelines. For women who participated in two competitions, individual differences in the positive effect of competition on cortisol and testosterone were consistent from one competition to another.	Participants were given sugar-free gum (Trident®, original flavor) to stimulate saliva production and a 20 ml polypropylene vial that they were asked to fill up to a marked 5 ml line on the side	sIgA	
Edwards, D. 2020 USA Observational study	Female	25 players	There was an increase in cortisol levels from baseline on the neutral day to the competition day, remaining virtually unchanged during the warm-up period and experiencing a significant increase during the actual competition period. Meanwhile, testosterone levels remained stable at baseline for the soccer players. There was an increase in estradiol levels from the neutral day to the competition day, intensifying during the warm-up but decreasing during the actual competition period.	Participants provided saliva samples exactly according to protocols published elsewhere. The samples were placed on ice, frozen within an hour, and stored (volleyball: −26 °C; soccer: −80 °C) until the assay.	Testosterone Cortisol	
Felisberto, P. 2022 Brazil			Concentrations of testosterone (high levels) combined with high levels of lean mass were associated with the technical performance and ball involvement of semi-professional soccer players.	Saliva was collected before and after the training match for subsequent measurement.	sIgA, IgG e IgM	
Figueiredo, P.  2019 Brazil Observational study	Male	18 Payers	External loads, such as variables derived from GPS, showed a stronger association with sIgA than with s-RPE (session rating of perceived exertion).	Participants received an oral fluid collector (OFC, IPRO Interactive, Oxfordshire, United Kingdom) containing a volume adequacy indicator with a clear color change when 0.5 mL (± 20%) is collected.	Cortisol Testosterone	
Filaire, E.2001 France Clinical trial	Male	17 players	Results indicate that, in the context of team soccer, a reduction in the testosterone-to-cortisol ratio does not necessarily result in an automatic decrease in team performance or a state of overtraining.	Soccer players provided three saliva samples upon waking (resting values, 8 AM), before breakfast (11:30 AM), and between 4 PM and 6 PM. Initial measurements were taken 1 day after the start of the season training (T1).	Testosterone sIgA Cortisol	
Filaire, E . 2003 France Cross-sectional study	Male	20 players	The levels of immune markers IgA, IgG, and IgM, along with hematological parameters, showed no changes. The subsequent reduction in performance coincided with modifications in team-specific mood states..	The soccer players provided two saliva samples upon waking (rest values, 8 am) and between 4:00 pm and 6:00 pm. The samples were stored in a freezer at −30 °C until analysis.	sIgA	
Fothergill, M. 2017 United Kingdom	Male age, 17.47, SD 0.64	18 players	The findings suggest that higher levels of stress are experienced by home team players in their home games.	The players were given Salicaps labels and were asked to produce about 2ml of saliva, which was collected through passive drooling. The samples were then stored at −20 °C within 8 h after collection and were analyzed within the recommended 28-day timeframe.	sIgA	
Francavilla, V. 2018 Italy Clinical Trial	Male	35 players	The results support the experimental use of salivary samples as a means to monitor changes in specific biomarkers, such as steroid hormones, in professional soccer players throughout a competitive season.	Saliva was collected using Salivette©devices (Sarstedt, Germany) through sterile cotton rolls placed in the mouth. Players were required to rinse their mouths with distilled water before starting the procedure. The Salivette©vials were stored in a refrigerator and delivered to the laboratory, where they were centrifuged at 1500 × g for 3 (min)–5 (max) minutes. The obtained liquid was volumetrically quantified and then transferred to low-binding polypropylene vials to be immediately frozen and stored at −20 °C.	Cortisol	
Fredericks, S. 2012 United Kingdon Clinical trial	18–39 years Male	 24 players	Overnight rest was sufficient for the recovery of mucosal IgA after training but not after two successive games, suggesting that sIgA can be used to monitor training in multi-sprint sports.	The athletes provided saliva samples approximately 10 min before the pre-game warm-up; post-game saliva samples were collected 10-15 min after the end of SM and OM. Participants refrained from consuming food and caffeine-containing products for at least 2 h before saliva collection. Unstimulated saliva was collected in sterile 15 ml centrifuge tubes for a period of 5 min and then stored at −80 °C until analyzed for SIgA concentration.	Micro RNAs (miRNAs)	
Freitas, C. 2016 Brazil Clinical trial	Male 15 years	26 players	The OM (official matches) led to a decrease in SIgA in young soccer players.	Participants in the saliva sample were required to chew on a simple (non-citrus acid) cotton salivate for 45 s. All saliva samples were subsequently labeled according to the subject, assay, and time and frozen at −80 °C.	Cortisol Testosterone	
Greig, M. 2006 UK Clinical trial	Age 24.7 ± 4.4 years	10 players	The physiological and mechanical responses were typically higher during the INT protocol (Intermittent Treadmill protocol specific to soccer) than during the SS protocol, tending to increase with exercise duration.	The participants in the saliva sample were required to chew on a simple (non-citrus, acidic) cotton salivate for 45 s. All saliva samples were subsequently labeled according to subject, assay, and time, and frozen at −80 °C.	Testosterone Cortisol	
Hicks, S. 2023 Clinical trial USA	Male	314 players	The levels of salivary miRNA accurately identified SRC (sports-related concussion) when not confounded by exercise.	The isolation from each saliva sample was not performed according to the manufacturer’s instructions with the miRNeasy kit (217084; Qiagen Inc., Germantown, MD, USA). The RNA quality was assessed using the Agilent 2100 Bioanalyzer (Agilent Technologies, Inc., Santa Clara, CA, USA). The TruSeq Small RNA Library Prep kit (RS-122-2001; Illumina, San Diego, CA, USA) was utilized to prepare RNA libraries.	Cortisol	
Jiménez, M. 2020 Spain Clinical trial	Male	95 players	An increase in testosterone was only observable when the team faced a real challenge, but it did not support a different response between the categories. Cortisol levels were lower in professional athletes (t =  − 3.456, p < 0.001) and semi-professionals (t =  − 4.400, p < 0.0001) than in amateurs (t =  − 2.835, p < 0.009)	Whole saliva samples were collected in Salivette® swab tubes (Sarstedt, Nümbrecht, Germany) and subsequently frozen at −40 °C.	Cortisol Testosterone	
Kargarfard, M. 2018 Iran Clinical trial	Male 24 years	30 players	Psychological overtraining scores were positively correlated with testosterone concentrations at 8 a.m. (r = 0.39; p = 0.015) and 5 p.m. (r = 0.37; p < 0.05), but negatively correlated with the T/C ratio at 8 a.m. (r =  − 0.38; p = 0.020).	Three samples of unstimulated saliva (2 ml each) were collected on rest days (8 a.m., 11 a.m., and 5 p.m.) from 30 male elite soccer players (age: 24.1 ± 3.8 years (mean ±SD)) and analyzed for cortisol and testosterone.	Cortisol Nitric Oxide	
Kayacan, Y. 2017 Turkey Clinical Trial	21 years Male	14 players	There were no statistically significant differences in the Cortisol Awakening Response (CAR) samples and during the competition. There were no correlations between cortisol parameters and the IDATE-T scores (P > 0.05).	To determine cortisol responses, saliva samples were collected on three different days during a regular league match.	Cortisol Desidroepiandrosterone	
Klentrou, P. 2016 Canada Clinical Trial	Male	26 players	There were no significant differences in baseline hormonal concentrations between trials or across weeks (p > 0.05). A significant time effect was found for testosterone (p = 0.02, [Formula: see text] = 0.14), which increased from pre-exercise to 5 min post-exercise in both resistance (27% ± 5%) and plyometric (12% ± 6%) protocols. Cortisol decreased similarly in both trials (p = 0.009, [Formula: see text] = 0.19) from baseline to post-control and then to 5 min post-exercise, following its typical circadian decrease during nighttime hours.	One milliliter of unstimulated whole saliva was collected from each participant using saliva swabs (Sarstedt Inc., Montreal, Que. Canada). Participants lightly moistened/chewed on the swab for 1 min. After sampling, the swabs were placed directly into plastic tubes. All saliva samples were transported in a cooled box and stored at −20 °C until analysis.	sIgA Cortisol	
Koerte, I. 2022 Germany Longitudinal Study	Male Ages 14 to 16 years	129 players	Until now, we have identified several panels of dysregulated miRNAs as primary outcomes for further investigation as potential diagnostic or prognostic biomarkers. After validating the primary hits, the data are prepared for bioinformatics analysis and prediction of genetic targets and dysregulated molecular pathways associated with RHI.	Players were required to ensure adequate hydration (they consumed 500 ml of water) due to the dehydration effects associated with reduced resting saliva flow rates, although hydration status was not recorded.	IGF-1 (Insulin-like Growth Factor 1) Interleukin 10	
La Fratta, I. 2021 Italy Observational Study	Male	56 players	Positive associations were found between cortisol, nitric oxide, feelings of anxiety, and competition outcomes. Significant differences were observed between winners and losers in cortisol (C) and oxytocin (OT) concentrations, with higher levels of OT in winners and higher levels of C in losers.	Saliva samples were collected using the Salivette system (Sarstedt Co., Nümbrecht, Germany), 30 min before the training session. Each sample was collected by having the participant place a swab under their tongue for 4 min. Saliva samples were stored at −80 °C until biochemical analysis.	Cortisol	
Labsy, Z. 2013 France Clinical trial	Male	9 players	The diurnal decline of both adrenal steroids was observed on both rest and exercise days in all conditions. There was a significant increase in salivary cortisol concentrations on morning exercise days and afternoon exercise days, respectively, at 12 p.m. (p < 0.05) and 4 p.m. (p < 0.01), compared to the other trials. This acute exercise response was not evident for DHEA.	Unstimulated saliva was collected by the participants themselves using Salitubes (DRG Diagnostics, Marburg, Germany). The Salitubes were promptly stored at 4 °C for one hour and at −20 °C for 3 days until analysis.	Cortisol Testosterone sIgA Alpha-amylase	
Lasisi, T. 2016 Nigeria Clinical trial	male 20 to 46 years old	22 players	These findings suggest that acute exercise caused a significant reduction in salivary flow rate, but no changes in levels of free IGF-1 (insulin-like growth factor 1) and IL-10 in saliva.	Samples of unstimulated whole saliva were collected by expectoration into sterile tubes before and after the soccer training session, and then centrifuged at 1,000 rpm for 2 min.	ACE I/D Polymorphism	
Lippi, G.2009 Italy Clinical trial	Male age: 27 ± 1 years	25 players	A highly significant correlation was observed between saliva and serum cortisol. The percentage of values above the upper limit of the reference range was almost identical for saliva and sérum.	Saliva was collected using Salivette devices (Sarstedt, Germany) with sterile cotton rolls placed in the mouth and then squeezed into pre-weighed containers.	Aldosterone Cortisone	
Lippi, G. 2016 Italy Clinical trial	Male	25 players	A significant correlation was observed between free testosterone in serum and saliva (r = 0.590; P = 0.002), while no significant correlation was found between total testosterone in serum and saliva (r = 0.181; P = 0.387). A significant correlation was found for the ratio of free testosterone to cortisol in serum and saliva (r = 0.43; P = 0.031).	  Saliva was also collected from each athlete using saliva sampling devices, which were validated for hormone measurement in saliva (Sali-Tube, DRG Instruments GmBH, Marburg, Germany).	Androstenedione (steroid hormone) Cortisol Testosterone	
Lopes, R. 2020 Clinical trial Portugal	Male	57 players	Negative correlations, age-controlled, were found between the occurrence of URS (upper respiratory symptoms) and sIgA, sAA, and sT.	Passive samples of unstimulated whole saliva were collected for 3 min with athletes sitting and their heads slightly tilted forward, using appropriate pre-weighed and pre-labeled 7mL plastic tubes (Sarstedt®), and immediately stored at approximately 4 °C in a polystyrene container with ice before transport and subsequent storage in the laboratory.	sIgA Cortisol	
Massidda, M. 2020 Japan Cross-sectional Study	Male ≥3 generations (n = 341, age 19.9 ± 5 years) ≥2 generations (n = 369, age 20.8 ± 1.4 years)	 710 players	The ACE I/D polymorphism was significantly associated with muscle injury using the dominant D model. The frequencies of DD+ID genotypes were significantly lower in injured groups than in non-injured groups with a low degree of heterogeneity. The ACE I/D polymorphism may influence susceptibility to the development of muscle injuries among soccer players.	A buccal swab was collected from each participant and stored in a tube with 1 ml of ethanol. Genomic DNA was extracted using a buccal swab according to the manufacturer’s instructions provided with a commercially available kit (Qiagen, Hilden, Germany). The concentration of extracted DNA was determined using a fluorometric method (via the Qubit from Invitrogen, Waltham, MA, USA).	Alpha-amylase Cortisol	
Maya, J. 2016 Chile Clinical trial	Female Age: 22.5 ± 2.1 years	22 players	Suggest that salivary concentrations of cortisol and testosterone increase especially after the first game of a final, without affecting IgA levels.	Unstimulated whole saliva (3 ml) was collected in a sterile 10 ml plastic bag and placed in a container with ice. The procedure lasted approximately 5 min in total. Saliva flow rate was not determined. The samples were later centrifuged at 1,500 g for 15 min, and the supernatant was stored frozen in microtubes at −20 °C until the samples were analyzed.	sIgA Cortisol	
McGuire, Amy;  2023 Ireland Clinical trial	Male 24 years	30 players	The decrease in s-IgA flow rate and s-IgA secretion rate were significantly associated with decreased LEA (low energy availability) in the preseason. Salivary testosterone significantly decreased from pre-games to the season but was significantly elevated post-games compared to the season. Salivary cortisol significantly decreased from pre-games to the season, remaining reduced post-games.	During saliva collection, players placed the oral fluid collection swab (Soma Bioscience, Wallingford, United Kingdom) on top of their tongue and closed their mouths until the swab collected 0.5 ml of fluid and turned blue. The swab was then placed in the OFC buffer vial for assays, following the manufacturer’s guidelines, sealed, and sent to a laboratory.	Testosterone	
McHale, T. 2016 USA Clinical trial	Male	26 players	Androstenedione increased significantly only during competition. Cortisol did not change with statistical significance during any of the conditions. Cortisol was the only hormonal measure that, with statistical significance, increased more during the match than during training. No statistical analysis was available for testosterone as all samples, except seven, were below the assay sensitivity (<3.0 pg/mL).	Individuals provided passive drool saliva samples 10 min before the start of the soccer training and 10 min before the warm-up period on the day of the soccer match.	sIgA Cortisol	
McHale, T. 2020 USA Observational Study	8 to 14 years old	84 players	Aldosterone levels increased during the soccer match and intra-squad game conditions, consistent with the view that aldosterone responds to physical stress. Cortisone increased during the soccer match and decreased during the intra-squad soccer training.	No report	Testosterone	
Mehdivand, A. 2010 Iran Clinical trial	Male 22 years	20 players	Cortisol, total protein, and salivary flow rate increased significantly. Differences in salivary IgA secretion rate were significant only in the defender group. The S-IgA/Pro (total protein) ratio in the defender group showed a significant difference compared to the defender and forward groups.	Before, immediately, and 24 h after exercise, unstimulated saliva samples were collected.	Cortisol sIgA	
Mehdivand, A.; 2011 Iran Cross-Sectional Study	Male 21 years	22 players	The results indicate that, compared to the start of the game, there were significant increases in salivary cortisol levels, total protein amount, and protein secretion rate after the match. Concurrently, the saliva flow rate, IgA secretion rate, and the value of the s-IgA/total protein ratio showed significant reductions during the same period (p < 0.05).	All participants underwent saliva collection before the game, at halftime, and immediately after the game. 	sIgA Cortisol	
Mehrsafar, A.2021 Iran Clinical trial	Male 26 to 48	90 players	Cognitive anxiety was a relevant predictor for the competitive response of cortisol and salivary alpha-amylase (sAA). Preliminary evidence that COVID-19 anxiety and competitive anxiety may have a negative impact on the athletic performance of professional soccer players during COVID-19 pandemic competitions.	Participants were instructed to drool saliva passively into a disposable plastic cup (2.5 mL) for 2 min. The cup contents were then transferred to polypropylene vials for storage at −20 °C until physiological assay.	Testosterone	
Minetto, M.2008 Italy Clinical Trial	Male	15 players	The smaller the decrease in performance, the greater the increase in cortisol associated with the training.	Saliva was collected from chewed cotton swabs (Sarstedt, Numbrecht, Germany). Saliva samples were centrifuged at 3000 RPM for 15 min at room temperature and stored at −20 °C.	Cortisol Testosterone	
Moreira, A. 2009 Brazil Clinical trial	Male Age: 23 ± 4 years	22 players	It has been reported that cortisol is elevated in players during a soccer match.	The subjects were seated, with their eyes open, head slightly tilted forward, and performing minimal orofacial movements. Unstimulated saliva was collected in sterile 15 ml centrifuge tubes for a period of 5 min. Immediately after collection, the saliva samples were frozen and stored at −80 °C until cortisol assay.	Circulating miRNAs neuroproteins (total/ phosphorylated tau associated with microtubules, neurofilament light polypeptide, ubiquitin C-terminal hydrolase-L1, glial fibrillary acidic protein) cytokines (IL-6, IL-10, TNF- α).	
Moreira, A. 2014 Brazil Clinical trial	Male 12.9 ± 0.2 years	34 players	A significant increase in the secretion rate of SIgA and a decrease in upper respiratory tract infection (URTI) symptoms were observed after the 2-week detraining period (p < 0.05). No changes were observed in cortisol levels during the study.	With the players seated and their heads slightly tilted forward, unstimulated saliva samples were collected through passive drooling into sterile 15 ml centrifuge tubes over a period of 5 min. The saliva samples were then stored at −80 °C until analyzed for cortisol and SIgA concentrations.	Cortisol Nitric Oxide	
Moreira, A. 2016 Brazil Clinical trial	Male 14.8 ± 0.4 years	16 players	Cumulative fatigue related to participation in a congested match schedule decreased testosterone concentration in young players and negatively affected their mucosal immunity and ability to perform certain technical actions.	All players remained seated, with eyes open, head slightly tilted forward, and making minimal orofacial movements during saliva collection. Unstimulated saliva was collected in a pre-weighed sterile 15 mL centrifuge tube over 5 min and stored at −80 °C until analysis.	sIgA	
Moreira, A. 2017 Brazil Clinical trial	14 years Male	16 players	Accumulated fatigue related to participation in a congested game schedule induced a decrease in testosterone concentration in young players and negatively affected their mucosal immunity and ability to perform certain technical actions.	Unstimulated saliva was collected in a pre-weighed sterile 15 mL centrifuge tube over 5 min and stored at −80 °C until analysis.	IGF-1 (Insulin-like Growth Factor 1); Interleukin 10	
Moreira, A. 2009 Brazil Clinical trial	Male	24 players	Variability in responses among players indicates the need to analyze results individually. Some athletes showed a decrease in s-IgA expressions.	Unstimulated saliva was collected in sterile 15 mL centrifuge tubes for a 5-minute period for each sample. Immediately after collection, saliva samples were frozen and stored at −80 °C until determination of s-IgA concentration.	Cortisol	
Moreira, A.;  2013 Brazil Clinical trial	Male 12 years	45 players	A significant difference was observed between the high and low testosterone groups regarding physical performance in pre-adolescent soccer players.	In a seated position with the head slightly tilted forward, unstimulated saliva samples were collected through passive drooling into sterile 15 ml centrifuge tubes for a period of 5 min. The saliva samples were then stored at −80 °C until tested for testosterone.	ACE I/D polymorphism	
Morgans, R. 2014 United Kingdon Clinical trial	Male Age: 26 ± 4 years	21 players	The congested schedule was sufficient to induce detectable disruptions to mucosal immunity in professional soccer players.	Participants received an oral fluid collector (OFC; IPRO Interactive, Oxfordshire, United Kingdom), which consists of a synthetic polymer material in a polypropylene tube.	sIgA Cortisol	
Morgans, R. 2022 Russia Clinical trial	Male Age: 26 ± 4 years	19 players	Mixed-model regressions revealed that longer playing time, total distance covered, and a higher number of high-intensity accelerations were associated with smaller differences in s-IgA between 30 min post-game and 60 min before kickoff, and between 60 min before kick-off and MD + 2. Additionally, greater distances of high intensity and sprint, and a higher number of high-intensity and maximum accelerations, were associated with smaller differences in cortisol between 60 min before kick-off and MD + 2.	The players were required to place an oral fluid collector (OFC II; Soma Bioscience, Oxfordshire, United Kingdom), which consists of a polymer-based swab attached to a polypropylene volume adequacy indicator rod, in their mouths. Participants were instructed to swallow any saliva present in the oral cavity before placing the collection device on the top of their tongue.	Alpha-amylase Cortisol	
Mortatti, A. 2012 Brazil Clinical trial	Male 19 years	14 players	There was observed a decrease in mucosal immunity, as measured by salivary IgA concentrations. Reductions in salivary IgA concentrations were noted during game 2 (r =  − 0.60; p < 0.05) and game 6 (r =  − 0.65; p < 0.05). No significant correlations were found between these variables in the remaining games played.	In a seated position with the head slightly tilted forward, unstimulated saliva samples were then collected through passive drooling into sterile 15 ml centrifuge tubes for a period of 5 min. The saliva samples were stored at −80 °C until they were analyzed for cortisol and IgA.	sIgA Cortisol	
Nakamura, D. 2006 Japan Clinical trial	Male	12 players	The SIgA level did not significantly decrease before the onset of symptoms of Upper Respiratory Tract Infections (URTI). However, the saliva flow rate and the SIgA secretion rate tended to decrease 3 days before the onset of URTI symptoms compared to the infection-free period (31.3+/-19, −42.2+/-20.6%, respectively).	There are no reports.	sIgA Cortisol	
Neave, N. 2003 United Kingdon Clinical trial	Male Age: 26.57 ± 4.39	17 players	Correlation between testosterone and changes in competitive performance.	The players were given labeled cups with lids and sugar-free gum, and were instructed to chew the gum and deposit enough saliva to fill the bottom of the cup (5 ml). Sample collection took 1 to 3 min. All samples were collected 1 h before the start of the games (16:45 h), and the samples were then frozen at −20 °C.	sIgA	
Nyakayiru, J. 2017 Netherlands Cross-sectional study	Male Age: 23 ± 1	32 players	Six days of supplementation with nitrate-rich beetroot juice improve high-intensity intermittent exercise performance in trained soccer players. The improvements in intermittent exercise performance were accompanied by a lower average heart rate during the high-intensity intermittent running test and were preceded by increases in plasma and salivary nitrate and nitrite concentrations.	  Saliva samples were collected in 2 mL Eppendorf tubes and stored at −80 °C until concentrations of nitrate and nitrite were determined in both saliva and plasma using chemiluminescence. The plasma samples were also stored at −80 °C for subsequent analysis of nitrate and nitrite concentrations	Testosterone	
Oliveira, T. 2009 Brazil Cross-sectional Study	Female 24 to 70 years old	32 players	The cortisol levels did not vary with the competition outcome. An early increase in circulating levels of both hormones (testosterone and cortisol) was detected before the match. Alongside hormonal responses, changes in mood and anxiety state were also observed between both teams, with more positive states in the winners and more negative states in the losers at the end of the match.	Saliva samples were collected to assess levels of testosterone (T) and cortisol (C) 30 min before the start of the match and before warm-up (pre-match sample) and 30 min after the end of the match (post-match sample). Saliva was provided by passive drool, without the use of any methods to stimulate salivation (e.g., gum).	Cortisol Testosterone	
Oliveira, T. 2009 Portugal Clinical trial	Female Ages: 24.24 ± 4.78 years	33 players	Players from both teams showed an early increase in both testosterone (T) and cortisol (C) levels before the game. Throughout the game, players from the winning team experienced an increase in T levels, while T levels decreased in the losers, resulting in significantly higher T levels in the winners than the losers at the end of the match. The change in T levels throughout the game was correlated with both observed performance and an increase in positive mood.	Saliva was provided through passive drooling, without the use of any methods to stimulate salivation (e.g., gum).	sIgA	
Owen, A. 2018 France Observational Study	Male Age: 24.9 ± 3.3 years	37 players	Significant correlations between s-IgA and training intensity were also observed. High-intensity soccer training sessions can lead to a significant decrease in s-IgA values during the post-exercise window compared to low-intensity (LI) sessions.	Collection kits IPRO OFC in combination with the real-time lateral flow device (LFD), respectively. Players were required to place a synthetic polymer-based swab attached to a volume adequacy indicator rod in their mouths. After the OFC kits collect 0.5 ml of oral fluid (collection times typically ranging from 20 to 50 s)	sIgA	
Peñailillo, L. 2015 Chile Clinical trial	Male 26 ± 3.5 years	9 players	The pre and post-game testosterone levels were correlated with post-game IgA concentrations. These results suggest that a soccer match induces catabolic stress, as indicated by the decrease in the T/C ratio. It appears that soccer players with smaller reductions in testosterone levels covered more distance and experienced less decline in their immune function.	Unstimulated whole saliva (3 ml) was collected in sterile plastic containers (10 ml) for approximately 5 min and placed in a container with ice. Subsequently, the samples were centrifuged at 1500 g for 15 min and stored frozen at −20 °C in Eppendorf tubes until the samples were analyzed.	Cortisol Testosterone sIgA	
Perciavalle, V. 2013 Italy Clinical trial	Male Age: 26 ± 3.5	18 players	The concentrations of testosterone and IgA decreased by 30.6% and 74.5%, respectively. The testosterone/cortisol ratio (T/C) decreased by 64.2% after the match. Changes in testosterone concentrations correlated with the distance covered. Pre and post-game testosterone levels correlated with post-game IgA concentrations. These results suggest that a soccer match induces catabolic stress, as indicated by the decrease in the T/C ratio. It appears that soccer players with smaller reductions in testosterone levels covered more distance and experienced less reduction in their immune function.	In summary, saliva samples (1 ml) were collected at rest in sterile containers and stored at −80 °C. Sugar-free gum (Vivident Xylit) was used to increase saliva flow in the participants.	Testosterone	
Perkins, E. 2022 United Kingdon Clinical trial	Male age: 27 ± 4	15 players	These findings confirm salivary IgA as a useful marker for the risk of URS (upper respiratory symptoms), but Epstein-Barr DNA was not. However, further research capturing a larger number of URS episodes is needed to fully determine the utility of this marker.	Provided weekly samples of unstimulated saliva (after a day of rest) and recorded URS. Saliva samples were analyzed for secretory IgA (ELISA) and Epstein-Barr DNA (qPCR).	sIgA	
Perroni, F.2023 Italy Clinical trial	Male Age: 13.5 ± 1.4 years	107 players	More specifically, T (testosterone) and C (cortisol) were shown to be differentially modulated in response to metabolic stress, and their ratio was reported as an indicator of anabolic/catabolic activity. This ratio is used in sports physiology as a marker for either or both overreaching and overtraining syndromes.	No reports.	Testosterone Cortisol 	
Pinto, J. 2021 Brazil Clinical trial	Male Age: 16.8 ± 0.5 years	20 players	There was a significant difference between the high and low salivary cortisol groups split post hoc in both matches (match 1: p < 0.001; ES: 2.7 and match 2: p < 0.001; ES: 2.3). The findings of this study indicate that the short recovery time did not influence the responses of salivary cortisol and internal load in the subsequent match.	The participants remained seated, with their eyes open, heads slightly tilted forward, making minimal orofacial movements to avoid stimulating salivary secretion. Immediately after collection, the tubes were stored on dry ice and transported to the laboratory, where they were stored at −80 °C until the time of analysis.	Cortisol	
Pintus, R. 2021 Italy Clinical trial	Male	21 players	Specifically, significant variations were observed for trimethylamine N-oxide, dimethylamine, hippuric acid, hypoxanthine, guanidoacetic acid, 3-hydroxybutyric acid, citric acid, and creatine.	No rreports	Trimethylamine N-oxide Dimethylamine hippuric acid hypoxanthine guanidoacetic acid 3-hydroxybutyric acid citric acid and creatine. 	
Pitti, E. 2019 Italy Clinical trial	Male	17 players	The salivary concentration of taurine increased after consecutive high-intensity exercises in male university soccer players.	A volume of 500 µL of the saliva sample was transferred to the filter device and centrifuged at 13, 800 × g at 4 °C for 90 min.	Total protein Taurine	
Ra, S. 2014 Japan Clinical trial	Male	37 players	There were no significant differences in any parameter between the two groups. Salivary flow rate and sIgA secretion rate significantly decreased after the program in the fatigue group. Additionally, salivary taurine concentration significantly increased after the program in the fatigue group. In the non-fatigue group, there were no significant differences in any parameter during the program	We also measured salivary concentrations of sIgA and taurine before and after the program.	sIgA Taurine	
Ranchordas, M. 2016 United Kingdom Case Study			There was a positive increase in salivary immunoglobulin-A (98 mg⋅ dl-1 to 135 mg⋅ dl-1), as well as a decline in the number of upper respiratory tract infection (URTI) symptoms (1.8 ± 2.0 vs. 0.25 ± 0.5 for weeks 0-4 and weeks 8-12, respectively).	No reports	sIgA	
Aceña, A. 2021 Costa Rica Clinical trial	Male	20 players	Moderate negative correlation between cortisol concentrations measured the day before the game and on the day of the game with the total accumulated distance in the microcycle, as well as moderate negative correlations between cortisol on the day before the match and the accumulated distance covered between 14-19 km/h (r =  − 0.32).	The analysis of cortisol concentration in saliva (nmol⋅ L-1) was performed within the first 30 min upon waking up, at 8 a.m., and before breakfast, without having brushed the teeth, considering that the variation in cortisol due to tooth brushing is close to 22% (unpublished data)	Cortisol	
Russell, M. 2017 United Kingdon Clinical trial	Male	14 players 	No differences were observed between the assays for other salivary markers (cortisol and testosterone/cortisol).	Saliva samples were collected in sterile containers (LabServe, New Zealand) using passive drool (∼2 ml over 2 min), which were then stored at −80 °C.	Testosterone Cortisol	
Santone, C. 2014 Italy Clinical trial	Male	14 players	Pre and pos t-test metabolomic analysis profiles show that it was possible to group athletes with better and worse performance, and the actual player’s role can be diagnosed by a different metabolite clustering profile	The collection was consistently done in the morning, three hours after breakfast. All saliva was pooled (totaling about 45 ml), and aliquots were taken for all validation tests.	Salivary Metabolomics	
Sari-Sarraf, V. 2006 Iran Clinical trial	Male	10 players	Performing specific soccer exercises at different times of the day did not affect the concentration of salivary IgA and the secretion rate or salivary cortisol in the short term. Two sessions of 90-minute exercises performed at moderate intensity with a 2.25-hour rest between them had no adverse effects on salivary IgA levels.	Unstimulated whole saliva was collected by expectoration for 5 min into pre-weighed sterile plastic containers (Sarstedt, United Kingdom) with the subject’s eyes open, head slightly tilted forward, and minimal orofacial movement. The samples were then aliquoted and stored at −80 °C for subsequent analysis.	sIgA Cortisol	
Sari-Sarraf, V. 2011 Iran Observational Study	Male	 10 players. 	The pattern of change in salivary responses, including solute secretion rate, IgA concentration, IgA secretion rate, IgA/osmolality ratio, cortisol, and cortisol secretion rate, did not differ between the two assays (p > 0.05).	Saliva samples were collected at four time points: before exercise attempts, immediately post-exercise, 24 h, and 48 h after exercise. Heart rate was monitored during exercise every 5 s; individuals assessed their thermal sensation (Tsens) and perceived exertion (PSE) every 15 min using the thermal sensation ] and Borg scales, respectively. Sterile pre-weighed plastic containers (Sarstedt, United Kingdom) were used for saliva collection.	sIgA	
Scheett, T. P. 1999 USA Cross-sectional study	Male Female	17 healthy children aged 8 to 11 years (4 females)	The cytokines increased after exercises consisting of a relatively brief 1.5-hour soccer practice in healthy children.	Salivary samples were collected using Salivettes (Sarstedt, Sparks, NV). Salivary and serum cortisol levels were determined by a commercial RIA (Diagnostic Products Corporation, Los Angeles, CA). The intra- and inter-assay coefficients of variation for this assay are 3.2% and 6.8%, respectively.	Cytokines –interleukin-(IL) 1beta (IL-1beta), IL-6 (IL-6), Cortisol	
Sparkes, W. 2020 United Kingdom Clinical trial	Male	12 players	Likely to very likely, small favorable responses occurred after the single session for testosterone (−15.2; ±6.1 pg/mL), cortisol (0.072; ±0.034 µg/dL), and testosterone/cortisol ratio (−96.6; ±36.7 AU) at +24 h compared to the double-session test. These data highlight that the performance of two training sessions in one day resulted in possible or very likely small impairments in neuromuscular performance, mood scores, and endocrine markers at +24 h compared to a single day of training session.	At all time points, 2 ml of saliva were collected via passive drool into sterile containers. Saliva samples were stored at −20 °C for seven days until the assay. After thawing and centrifugation (2,000 rotations per minute ×10 min), saliva samples were analyzed in duplicate for testosterone and cortisol concentrations using commercial kits (Salimetrics LLC, USA).	Testosterone Cortisol	
Sparkes, W. 2022 United Kingdom Clinical trial	Male age, 21 ± 2 years;	12 players	Movement demands were monitored through Global Positioning Systems (GPS), counter-movement jump (CMJ), saliva (testosterone and cortisol), and a brief assessment of mood (BAM+) collected immediately before and after Small-Sided Games (SSG) training. The results suggest that CMJ variables and hormonal markers have good reliability between weeks when measuring athletes at rest (CV, 2.1-7.7%; ICC: 0.82–0.98), but BAM+ does not (CV, 23.5%; ICC: 0.47). GPS variables showed low to high repeatability during SSG training, with reliability statistics varying among metrics (CV, 4.4–62.4%; ICC: 0.30–0.81).	Movement demands were monitored through Global Positioning Systems (GPS), counter-movement jump (CMJ), saliva (testosterone and cortisol), and a brief assessment of mood (BAM+) collected immediately before and after Small-Sided Games (SSG) training.	Cortisol Testosterone	
Springham, M. 2021 United Kingdom Longitudinal	Male	18 players	The analysis identified a slight (P = 0.003) cross-suppression of salivary immunoglobulin A, minor reductions in salivary alpha-amylase (P = 0.047), and salivary cortisol (P = 0.007), and trivial changes in salivary testosterone (P > 0.05). The testosterone/cortisol ratio typically responded inversely to changes in the player’s workload	The players were asked to sit quietly, swallow the existing saliva in their mouths, and then place an oral fluid collector (OFC; SOMA Bioscience, Wallingford, United Kingdom) on their tongue. With their mouths closed, 0.5 mL of saliva was collected, as indicated by the volume adequacy indicator on the OFC. The OFC was then placed in 3 mL of buffer solution in a custom 10 mL container (OFC buffer; SOMA Bioscience) and gently hand-mixed for 2 min.	Testosterone Cortisol sIgA Alpha-amylase	
Springham, M. 2022 United Kingdom Longitudinal	Male	18 players	Testosterone had non-linear relationships with chronic total running distance (P = 0.015; Cohen’s D: large), high metabolic load (P = 0.001; small), and high-speed running (P = 0.001; trivial) and linear relationships with chronic sRPE (P = 0.002; moderate ↓) and acute:chronic high-speed running distance (P = 0.001; trivial ↑). Cortisol had a non-linear relationship with chronic high-speed running distance (P = 0.001; trivial). Testosterone:cortisol had non-linear relationships with chronic decelerations (P = 0.039; small) and chronic sum of acceleration and deceleration load (P = 0.039; small). Non-linear relationships typically indicated optimal hormonal responses in moderate squad loads. No load variable clearly related to changes in salivary immunoglobulin A or alpha-amylase.	They were instructed to sit calmly, swallow the existing saliva in their mouths, and then place an oral fluid collector (OFC; SOMA Bioscience, Wallingford, United Kingdom) on their tongue. With their mouths closed, 0.5 ml of saliva was collected, as indicated by the volume adequacy indicator on the OFC. The OFC was then placed in 3 ml of buffer solution in a custom 10 ml container (OFC buffer; SOMA Bioscience, Wallingford, United Kingdom) and gently hand-mixed for 2 min	Cortisol Testosterone sIgA Alpha-Amylase	
Starzak, D. 2016 South Africa Clinical trial	11 to 13 years Female Male	34 players	12 weeks of specific soccer training increased mucosal immunity and body composition.	Saliva samples were collected between 7:30 and 8:30, approximately 90 min after waking up.	sIgA Alpha-amylase	
Thorpe, R.2012 United Kingdon Clinical trial	Male	7 players	No differences were observed from pre-game to post-game in the testosterone to cortisol ratio, immunoglobulin (Ig) A, IgM, or IgG.	Unstimulated whole saliva was collected by expectoration for 5 min into pre-weighed sterile plastic containers (Sarstedt, Nümbrecht, Germany). All saliva samples were placed on ice before being frozen at −80 °C until analysis.	Cortisol Testosterone Creatine Kinase (CK) Myoglobin (MYO) sIgA IgM IgG	
Vardiman, JP. 2011 USA Clinical trial	Female	12 players	A wide range of measurable levels of s-IgA at different time points for athletes and controls and the lack of a relationship between s-IgA levels and TSDs (total symptom days).	Saliva samples were collected within 1 h of the same time of day throughout the sports training to avoid diurnal effects. Saliva sample collection occurred exactly at the same time of day for the control group.	sIgA	
Zanetti, V. 2021 Brazil Clinical trial	Male	25 players.	The authors did not report a significant difference in salivary testosterone concentration, anthropometric measurements, aerobic capacity, and counter-movement and vertical jump performances at the beginning of the season between starting players and substitutes.	Unstimulated saliva samples were collected using the passive drool method into sterile 15 ml centrifuge tubes over a period of 5 min. The saliva samples were then stored at −80 °C until analyzed for testosterone concentration.	Testosterone	
Notes.

Akimoto et al. (2003); Alzahrani et al. (2019); Alzharani et al. (2020); Amato et al. (2018); Arruda et al. (2015); Arruda et al. (2016); Askari et al. (2011); Azarbayjani et al. (2011); Baldari et al. (2009); Baralic et al. (2015); Bishop et al. (1999); Bonato et al. (2020); Broodryk et al. (2021); Broodryk et al. (2017); Broodryk et al. (2020); Casto & Edwards (2016); Castro-Sepulveda et al. (2018); D’Ercole et al. (2013); Rodrigues De Araujo et al. (2018); Di Luigi et al. (2006); Dionísio et al. (2017); Edwards & Casto (2015); Edwards, Wetzel & Wyner (2006); Edwards & Casto (2013); Edwards & Turan (2020); Felisberto et al. (2022); Figueiredo, Nassis & Brito (2019); Filaire et al. (2001); Filaire, Lac & Pequignot (2003); Fothergill, Wolfson & Neave (2017); Francavilla et al. (2018); Fredericks et al. (2012); Freitas et al. (2016); Greig, McNaughton & Lovell (2006); Hicks et al. (2023); Jiménez et al. (2020); Kargarfard et al. (2018); Kayacan et al. (2017); Klentrou et al. (2016); Koerte et al. (2022); La Fratta et al. (2021); Labsy et al. (2013); Lasisi & Adeniyi (2016); Lippi et al. (2009); Lippi et al. (2016); Lopes et al. (2020); Massidda et al. (2020); Maya et al. (2016); McGuire, Warrington & Doyle (2023); McHale et al. (2016); McHale et al. (2020); Mehdivand et al. (2010); Mehrsafar et al. (2021); Minetto et al. (2008); Moreira et al. (2009a); Moreira et al. (2009b); Moreira et al. (2014); Moreira et al. (2016); Moreira et al. (2017); Moreira et al. (2013); Morgans et al. (2014); Morgans et al. (2022); Mortatti et al. (2012); Nakamura et al. (2006); Neave & Wolfson (2003); Nyakayiru et al. (2017); Oliveira, Gouveia & Oliveira (2009); Owen et al. (2018); Peñailillo et al. (2015); Perciavalle et al. (2013); Perkins & Davison (2022); Perroni et al. (2023); Pinto et al. (2021); Pintus et al. (2021); Pitti et al. (2019); Ra et al. (2014a); Ra et al. (2014b); Ranchordas, Bannock & Robinson (2016); Aceńa (2021); Russell et al. (2017); Santone et al. (2014); Sari-Sarraf, Reilly & Doran (2006); Sari-Sarraf et al. (2011); Scheett et al. (1999); Sparkes et al. (2020); Sparkes et al. (2022); Springham et al. (2021); Springham et al. (2022); Starzak, Konkol & Mckune (2016); Thorpe & Sunderland (2012); Vardiman et al. (2011); Zanetti et al. (2021).

The studies were conducted between 1999 and 2023, as follows: two published in 1999, one in 2001, three in 2003, four in 2006, one in 2007, one in 2008, six in 2009, one in 2010, five in 2011, three in 2012, five in 2013, four in 2014, four in 2015, 11 in 2016, seven in 2017, six in 2018, three in 2019, nine in 2020, eight in 2021, six in 2022, and three in 2023 (Fig. 2).

Figure 2 Number of publications according to year: (n = 2) in 1999, (n = 1) in 2001, (n = 3) in 2003, (n = 4) in 2006, (n = 1) in 2007, (n = 1) in 2008,(n = 6) in 2009,(n = 1) in 2010, (n = 5) in 2011, (n = 3) in 2012, (n = 5) in 2013, (n = 4) in 2014, (n = 4) in 2015, (n = 12) in 2016, (n = 7) in 2017, (n = 6) in 2018, (n = 3) in 2019, (n = 9) in 2020, (n = 8) in 2021, (n = 6) in 2022 and (n = 3) in 2023.

The included studies showed a total of 3,744 individual participants. Most of these studies were performed with male individuals (84.04%, n = 78 studies, with a total of 3,132 participants), whereas those developed with female individuals were 15.96% (n = 15 studies, with a total of 612 participants) and 3.5% of studies were performed with male/female individuals (n = 5, with a total of 145 participants).

Concerning the most studied salivary biomarkers in soccer, cortisol (n = 53, 33.2%) was the most evaluated marker, followed by testosterone (n = 35, 21.8%), salivary IgA (n = 33, 20.6%), salivary alpha-amylase (n = 7, 4.4%), genetic polymorphisms (n = 4, 2.5%) and miRNAs (n = 2, 1.2%), (Fig. 3).

Figure 3 Salivary biomarkers: cortisol (n = 53), testosterone (n = 35), salivary IgA (n = 33), salivary alpha-amylase (n = 7), MiRNAs (n = 2), genetic polymorphisms (n = 4), and others (n = 26).

Among the countries where most studies were conducted are Brazil (n = 18), Italy (n = 14), United States (n = 12), United Kingdom (n = 10), Iran (n = 8), Australia (n = 4), Japan (n = 4), France (n = 3), Chile (n = 3), Spain (n = 2), South Africa (n = 2), Portugal (n = 2), Nigeria (n = 1), Sweden (n = 1), Saudi Arabia (n = 1), Serbia (n = 1), Turkey (n = 1), Canada (n = 1), Germany (n = 1), Netherlands (n = 1), China (n = 1), Costa Rica (n = 1) and New Zealand (n = 1).

Regarding the type of saliva collection, 39 studies used unstimulated saliva, 24 studies employed some form of salivary collector, 20 studies utilized stimulated saliva, and 10 studies did not report the saliva collection method (Fig. 4).

Figure 4 Type of saliva collection.

Type of saliva collection: studies used stimulated whole saliva ( n = 20), studies used unstimulated whole saliva (n = 39), studies used some type of salivary collector (n = 24) and not report (n = 10).

Discussion

In this scoping review, we mapped the scientific literature on the currently available procedures for saliva collection and analysis in soccer players. The most studied salivary biomarkers were cortisol (n = 53), testosterone (n = 35), salivary IgA (n = 33), salivary alpha-amylase (n = 7), genetic polymorphisms (n = 4) and miRNAs (n = 2).

Understanding biomarkers in the sports setting, notably cortisol, testosterone, salivary IgA, and salivary alpha-amylase, is crucial for the comprehensive assessment of athletes’ performance, recovery, and physiological status (D’Ercole et al., 2013; Lopes et al., 2020). These biomarkers play fundamental roles in athletes’ health and performance, with each providing valuable insights.

Cortisol was the most evaluated biomarker in the articles included in this review, monitored in 53 studies. In the sports context, cortisol levels serve as sensitive indicators of the physical and psychological stress experienced by athletes, as evidenced by previous studies (Moreira et al., 2009b; Moreira et al., 2009a; Morgans et al., 2022). Elevated concentrations of this hormone may signal overtraining or chronic stress, potentially compromising both performance and athletes’ health. Assessing cortisol levels in saliva stands out as an effective approach, providing a practical and less invasive alternative to blood measurement in soccer. Nevertheless, studies have revealed significant correlations between salivary and blood concentrations of cortisol at rest. During submaximal exercises, the association between salivary and blood cortisol levels indicated a joint response to increased blood lactate. After exercise, significant correlations persisted, especially in intense protocols, such as the 30-second Wingate test and competitive training matches (Lippi et al., 2009).

Delving into the salivary concentrations of testosterone, an incontrovertible nexus emerges between the salivary testosterone levels and the physical prowess exhibited by soccer athletes. This confluence yields invaluable insights, facilitating the refinement of training regimens, prophylactic measures against injuries, and the enhancement of athletic performance. However, it should be taken into account that testosterone levels can vary due to numerous factors, such as age, sex, menstrual cycle, stress, and diet, which can act as confounding variables (Borchers et al., 2022).

Soccer involves intense physical activity and strenuous training, both of which have significant impacts on the immune function of athletes. Studies indicate an association between high-intensity and long-duration exercises and a temporary weakening of the immune system, increasing susceptibility to upper respiratory tract infections (Cadegiani, Kater & Gazola, 2019). A valuable indicator of this dynamic is the measurement of salivary IgA, assessed in 33 studies in this review. This assessment not only reflects the mucosal immune function but also highlights the response to training-induced stress (Filaire, Lac & Pequignot, 2003). Decreased levels of salivary IgA have been correlated with a higher risk of respiratory infections in athletes, emphasizing the importance of monitoring these levels for assessing athletes’ immune status. This approach allows the identification of periods of increased vulnerability to infections, representing a valuable tool in managing the health of high-performance athletes (Cadegiani, Kater & Gazola, 2019).

Salivary alpha-amylase emerges as a crucial biomarker in the sports setting, related to stress, as discussed in previous studies (Lopes et al., 2020; Mehrsafar et al., 2021). During physical exercise, especially in high-intensity situations, stimulation of sympathetic activity results in changes in salivary alpha-amylase levels, thus offering a unique window to assess athletes’ physiological response in situations of physical stress. Salivary amylase was evaluated in seven articles included in this review.

The relationship between genetic polymorphism and sport performance is a promising area of investigation. For instance, analysis of genetic data and the identification of variants has can be widely assessed in the oral fluid. Research on genetic polymorphisms in saliva represents a growing field in sports, providing valuable insights into athletes’ individual response to training and competitive demands (Massidda et al., 2020). This assessment was conducted in four studies with soccer players. Understanding these genetic nuances allows a more personalized approach in training and sports nutrition, optimizing strategies to meet the specific needs of each athlete (Hall et al., 2022). However, it is crucial to conduct further research to fully understand how genetic polymorphisms can impact sports performance and, more importantly, how this information can be applied ethically and effectively in the sports environment (Dionísio et al., 2017).

Still underexplored, microRNAs (miRNAs), small molecules of ribonucleic acid (RNA), emerge as protagonists in gene expression regulation, being studied in various areas of biology, including exercise physiology and sports performance, such as in soccer. Although the direct relationship between miRNAs and sports performance is still being explored, there are indications suggesting that some miRNAs may influence characteristics such as muscle strength and endurance. The study of these molecules also extends to injury prevention and prediction, highlighting their potential importance in athletic health management (Hicks et al., 2023; Koerte et al., 2022; Wyczechowska et al., 2023).

In the realm of soccer, research on cytokines primarily focuses on analyzing the inflammatory response, post-exercise recovery, and potential impacts on injuries. Intense sports practice, such as soccer, can trigger muscle microinjuries, initiating an inflammatory response in which cytokines such as interleukins (IL-6, IL-10) and tumor necrosis factor-alpha (TNF-alpha) play essential roles. The evaluation of these cytokines becomes a valuable tool for monitoring the physiological state of athletes, facilitating the adaptation of training and recovery protocols. However, it is crucial to adopt an integrated approach when interpreting cytokine levels, considering factors such as training intensity, season stage, individual health status, and others (Koerte et al., 2022). Interleukins (ILs), a group of proteins involved in communication between immune system cells, have been studied in the context of soccer and sports in general. Exploring their relationship with the inflammatory response, post-exercise recovery, and potential implications for athletes’ health highlights their crucial role (Scheett et al., 1999).

The meticulously conducted research involving salivary biomarkers in athletes, coupled with well-defined control groups, holds the potential for extrapolation to athletes’ everyday lives and their clubs, particularly when longitudinal studies are validated. Pre-, trans-, and post-competition assessments, along with monitoring during training sessions, can provide invaluable insights into the development of monitoring tools. This integration of clinical aspects with sports training fosters the optimization of athlete preparation, recovery, and performance. Such endeavors extend beyond soccer, encompassing all sports disciplines seeking advancement in athletic prowess.

The use of different saliva collection methods in soccer players was observed, with the most used being stimulated saliva collection, unstimulated saliva collection, and the use of specific salivary collectors such as salivette and swabs. Stimulated saliva collection, in which participants are instructed to chew a stimulus, such as an unflavored chewing gum or parafilm, has been frequently used due to its ability to provide larger saliva sample volumes (Al Habobe et al., 2024). On the other hand, unstimulated saliva collection also plays a significant role, offering a more natural and less intrusive approach. This method is particularly useful in situations where stimulated collection may not be practical, such as during soccer matches. Unstimulated collection allows obtaining saliva samples under conditions closer to those experienced by athletes during the game, potentially reflecting more accurately the impact of physical and psychological stress on performance (Thorpe & Sunderland, 2012). As any other body fluid that can be used in diagnostic approach, saliva requires proper collection/sampling methods, precise sampling time, appropriate handling and transportation conditions, and eventually, established storage considerations until further analysis of samples (Mortazavi, Yousefi-Koma & Yousefi-Koma, 2024). In this regard, it is paramount to highlight that not only the type of saliva collected (whole saliva or gland-specific) but also the circadian rhythm (Dawes, 1972) as well as the presence or absence of stimuli (Navazesh & Kumar, 2008) can impact the reliability and the variability in findings, subsequently affecting the interpretation of results obtained in the assessment of salivary biomarkers. In this sense, since there are different ways of collecting saliva samples, the choice and standardization of a method to perform it can enhance the significance of the findings. For instance, a recent study demonstrated that the results obtained from a diverse group of salivary biomarkers assessed by both unstimulated and chew-stimulated saliva were similar, but not the oral rinse method due to dilution of the saliva sample (Al Habobe et al., 2024). Until now, there is no consensus regarding the use or not of saliva sampling devices, such as Salivette®, Parafilm®, and paraffin wax since some authors argue that the use of non-stimulated saliva offers an unmanipulated/authentic sample (Eventov-Friedman et al., 2019), whilst others declare that the use of some devices decreases the presence of interferes, which leads to obtain faster and more accurate results (Jamieson et al., 2020; Mortazavi, Yousefi-Koma & Yousefi-Koma, 2024). Thus, it is clear that a guideline that provides essential methodological details to standardize the use of saliva in different contexts is widely necessary.

The results of the studies indicated beneficial effects in monitoring salivary biomarkers in the assessment of sports performance, although most studies did not include a comparable control group. The inclusion of control groups in studies involving human subjects is paramount for ensuring internal validity, thus facilitating valid comparisons by isolating specific effects of the studied variable and mitigating confounding variables that might distort results. Different types of control groups can be considered for such studies, such as a placebo control group, facilitating the observation of cause and effect; an active control group, where athletes and non-athletes’ marker levels could be compared; or a historical control group, comparing data from existing non-athlete datasets, thereby expediting the study process. The incorporation of control groups, as described in this review, is essential for ensuring its validity, reliability, and interpretability, thereby contributing to significant advancements in related fields.

Besides, it is utmost of importance to point out that the integration of different biomarkers and the understanding of the relationships between their salivary and blood manifestations not only enhance the assessment of athletes’ physiological state but also provide a solid foundation for implementing personalized training and recovery strategies, optimizing sports performance in a holistic manner. Therefore, it is worth mentioning that the improvement of understanding about the relationships of biomarkers assessed both in saliva and blood is associated not only with type but also with specific and standard analytical techniques that should be used to assess these biomarkers.

Among the limitations encountered, many studies fail to report the methodologies for saliva analysis and collection, a critical aspect necessitating attention in future research endeavors. Additionally, there is a pressing need for the establishment of appropriate control groups tailored to the profile of the studied population, such as reference values for physically active individuals (Borchers et al., 2022), as well as the identification of yet unstudied biomarkers and an understanding of the physiological response’s complexity concerning lifestyle and the specific sporting discipline practiced, for instance. Such methodological intricacies are imperative for validating the undertaken approaches. The lack of standardization of saliva collection and processing methods needs to be addressed (Mortazavi, Yousefi-Koma & Yousefi-Koma, 2024) since it significantly influences the obtained results, as recently evidenced by our research group (Eduardo et al., 2022). We are currently formulating guidelines to guide future studies in collaboration with researchers worldwide who share an interest in this niche area of investigation.

Conclusions

In conclusion, this scoping review provides a comprehensive overview of the current landscape of salivary biomarker research in soccer. The findings underscore the importance of these biomarkers in assessing athletes’ physiological responses and overall well-being. Future research should focus on refining methodologies, exploring additional biomarkers, and investigating the practical implications of salivary biomarker monitoring in soccer and other sports.

Additional Information and Declarations

Competing Interests

Author Contributions

Data Availability

Manuel Jimenez is an Academic Editor for PeerJ.

Joyce Ferreira performed the experiments, analyzed the data, prepared figures and/or tables, and approved the final draft.

Manuel Jimenez conceived and designed the experiments, analyzed the data, authored or reviewed drafts of the article, and approved the final draft.

Agatha Cerqueira performed the experiments, analyzed the data, prepared figures and/or tables, and approved the final draft.

Joana Rodrigues da Silva performed the experiments, analyzed the data, prepared figures and/or tables, and approved the final draft.

Bruno Souza analyzed the data, prepared figures and/or tables, authored or reviewed drafts of the article, and approved the final draft.

Lucas Berard performed the experiments, analyzed the data, prepared figures and/or tables, and approved the final draft.

Andre L.L. Bachi conceived and designed the experiments, analyzed the data, authored or reviewed drafts of the article, and approved the final draft.

Naile Dame-Teixeira conceived and designed the experiments, analyzed the data, authored or reviewed drafts of the article, and approved the final draft.

Neide Coto conceived and designed the experiments, analyzed the data, authored or reviewed drafts of the article, and approved the final draft.

Debora Heller conceived and designed the experiments, analyzed the data, prepared figures and/or tables, authored or reviewed drafts of the article, and approved the final draft.

The following information was supplied regarding data availability:

This is a literature review.

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
