# Peer review of "Saliva as a diagnostic tool in soccer: a scoping review"

_PeerJ, doi:10.7717/peerj.18032_

## Round 0.1 · original submission · Minor Revisions

Please address the comments raised by the external reviewers.

Reviewer 1 ·

Basic reporting

The article is well-written and clear in its presentation. The language used is professional and unambiguous, making it accessible to a broad audience. The structure conforms to the standards of the journal, with a logical organization of the content into coherent paragraphs and subsections. The references are well-referenced and relevant to the topic. The article meets the criteria for broad and cross-disciplinary interest and is within the scope of the journal.

Experimental design

The study design is sound, with a clear research question and a comprehensive literature search. The inclusion and exclusion criteria are well-defined, and the selection process is transparent. The data extraction and synthesis are thorough, and the results are presented in a clear and concise manner. However, the article could benefit from a more detailed description of the study's limitations and potential biases.

Validity of the findings

The findings of the study are valid and contribute to the current understanding of the use of saliva as a diagnostic tool in soccer. The results are well-supported by the literature and provide a comprehensive overview of the current landscape of salivary biomarker research in soccer. However, the article could benefit from a more detailed discussion of the implications of the findings and potential future directions for research

Additional comments

General comments

Dear Authors,
Your manuscript “Saliva as a Diagnostic Tool in Soccer: A Scoping Review” presents a compelling overview of the current state of research.

To further strengthen your review and its impact on the field, I recommend the following enhancements:

1. Standardization of Protocols: Highlight the necessity for standardized saliva collection and analysis protocols. Propose a set of guidelines or reference a consensus in the field to aid future researchers in aligning their methodologies.

2. Comprehensive Methodological Reporting: Encourage detailed reporting of methodologies in future studies. Perhaps include a checklist or template in the appendix to guide researchers in providing essential methodological details.

3. Control Group Inclusion: Advocate more strongly for the use of control groups. Discuss the types of control groups that could be employed and how they would enhance the interpretability of the results.

4. Correlation Between Biomarkers: Emphasize the importance of understanding the relationship between salivary and blood biomarkers. Suggest specific analytical techniques or study designs that could shed light on these correlations.

5. Genetic Polymorphism Research: Urge for in-depth exploration of genetic polymorphisms. Outline potential research questions or study designs that could illuminate the role of genetics in salivary biomarker variability.

6. Practical Implications: Stress the need for research with clear practical applications. Encourage studies that not only explore biomarkers but also test their utility in real-world sports settings.

7. Limitations and Future Directions: Expand the discussion on limitations and future research. Offer a vision for how the field might evolve and what key questions need to be addressed.

By incorporating these points, your review will not only provide a snapshot of the current research landscape but also chart a course for future investigations that can significantly advance our understanding and application of salivary diagnostics in sports performance.

Cite this review as

·

Basic reporting

Background should explain what the problem is and explain why saliva was chosen.mments

Experimental design

1. In the Introduction of this manuscript, we cannot find out what the problem of this study is. It should be explained in the introduction, at present what techniques are the gold standard for analyzing athletic performance in soccer, and what are the shortcomings of these techniques so that saliva is the best alternative to overcome the shortcomings of previous techniques.
2. The author should explain why this review is discussing soccer, not other sports.

Validity of the findings

1. In manuscript the authors wrote: The most studied salivary biomarkers were 162 cortisol (n=59), testosterone (n=42), salivary IgA (n=45, 23%), Creatine MiRNAs (n=4), salivary163 alpha-amylase (n=8), and genetic polymorphisms (n=5).
But we could not find discussion/analysis about testosterone and genetic polymorphisms in Discussion section.
2. The aim of this study was:
Therefore, here we aimed to map the literature on using saliva as a diagnostic tool in soccer, analyzing which salivary biomarkers are employed and describing the available protocols.

In this manuscript, the authors did not mention the saliva analysis method, one of important point for protocols

3. The authors should discuss which is the best saliva collection method between stimulated, unstimulated, and the use of specific salivary collectors such as salivette and swabs for soccer players

Additional comments

-

---

## Round 0.2 · accepted · Accept

Congratulations and thank you for addressing and revising the paper as per reviewers.